# Performance Status Evaluation of an Electric Vehicle Charging Infrastructure Based on the Fuzzy Comprehensive Evaluation Method

**Qiushuo Li [1], Yong Xiao [1], Shuaishuai Zhao [2], Xianwen Zhu [1], Zongyi Wang [3], Zisheng Liu [2], Ling Wang [2], Xiangwu Yan [2,\*] and Yan Wang [1]**

[1]    China Southern Power Grid Science Research Institute Co., Ltd., Guangzhou 510080, China;
       liqs@csg.cn (Q.L.); xiaoyong@csg.cn (Y.X.); zhuxw1@csg.cn (X.Z.); wangyan2@csg.cn (Y.W.)
[2]    North China Electric Power University, Baoding 071003, China; zhao2017shuaishuai@163.com (S.Z.);
       louisancepu@163.com (Z.L.); wlCatherine@163.com (L.W.)
[3]    China Southern Power Grid Company Limited, Guangzhou 510623, China; wangzongyi@csg.cn
\*    Correspondence: xiangwuy@ncepu.edu.cn; Tel.: +86-0312-7522862

**Abstract:** Performance status evaluation is essential for the safe running of electric vehicle (EV) charging infrastructure. With the development of the EV industry, the EV charging infrastructure industry has advanced considerably. Safe and reliable operation of the charging infrastructure is important for the development of EVs. As such, we propose a comprehensive evaluation method to assess the performance condition of an EV charging infrastructure. First, based on the analysis of the existing EV charging principles, we established an evaluation index system for EV charging infrastructure. Second, the subjective weight, objective weight, and comprehensive weight of the index system were determined through analytic hierarchy processes (AHP) and the entropy weight method. Then, we used fuzzy comprehensive evaluation to appraise the performance of the charging infrastructure through expert investigation. Finally, based on the actual data from an EV charger, the performance conditions of the EV charging infrastructure were evaluated to demonstrate the feasibility of the method and the reliability of the index system.

**Keywords:** electric vehicle charging infrastructure; fuzzy comprehensive evaluation; performance evaluation

---

## 1. Introduction

With the intensification of the fossil energy crisis and the deteriorating ecological environment, it is necessary to replace fuel-based vehicles with alternative energy sources to reduce air pollution. Electric vehicles (EVs) convert electrical energy into mechanical energy, which achieved zero emissions of automobile exhaust. EVs have become a new generation of transportation tools for many people. The EV development prospects are broad and a trend in replacing fuel-based vehicles has been observed. An increase in the number of EVs requires an increase in the number of EV charging stations and greater EV charging infrastructure. Statistics show that the number of alternative energy vehicles in China reached 2.61 million as of 2018. According to the Chinese Electric Vehicle Charging Infrastructure Promotion Alliance, as of January 2019, the cumulative number of electric vehicle charging stations in the country is 0.853 million. Following the standard of "one car and one pile", much room remains for the development of the number of charging piles (stations) for EVs [1]. Therefore, a safe, reliable, and convenient operation mode for EVs would form the basis for maintaining their sustainable development [2,3]. Knowing the fault characteristics and having a protection control strategy for

EV charging would be important for the safe operation of the distribution network of the charging equipment [4].

Faced with a rapid growth in the number of EVs, the development level of fault assessment and diagnostic technology for EV charging facilities in China is still low. As a result, a few safety-related EV charging accidents have occurred. The safety of the charging equipment is related to the promotion of EVs, which might lead to major accidents involving personal safety and property damage. Therefore, research on the performance evaluation and safety warnings for the charging equipment is necessary [5].

Some work has been completed on the safety of charging equipment. In Bouabana et al. [6], a testing device was developed that could analyze a new charging station by means of a power test and a security test. Li et al. [7] designed an early warning model for EV charging security to decrease the safety-related accidents involving EV charging. Kim et al. [8] studied the methodological evaluation and testing standards for on-board chargers and provided some suggestions for the test method.

Constructing an effective evaluation index system and improving the evaluation efficiency of the operating charger are some of the basic aspects necessary to ensure the safe and reliable operation of an EV charger. A study established a performance evaluation system for the drive motor and verified the integrity and operability of the system [9]. The authors showed that it is necessary to construct an evaluation system for the evaluation of the equipment. Another study focused on EV charging safety and analyzed the factors affecting the charging safety of EVs with regards to the insulation level of the charging equipment, the charging environment, and the protection measures of the charging facilities. The authors enumerated the common charging safety hazards [10]. In Ye et al. [11], an evaluation index model of the charger was built, considering both the electrical and safety performance, to evaluate the performance of the charger. In Wang et al. [12], by analyzing the electrical performance and safety performance of the EV chargers, the electrical and safety performance of the charger was studied using the fuzzy analytic hierarchy process (AHP). As the current development of the charging equipment industry is still immature, regional differences exist in standards, which complicates establishing procedures for unified management and evaluation of the charging equipment.

In this paper, based on the actual operation and maintenance of the site, the system indicating the characteristics of the charger was divided into three parts: General performance, electrical performance, and safety performance.

Selecting a reasonable and efficient evaluation method can help with the timely identification of faults in a charger and can increase the accuracy and efficiency of the infrastructure operation and maintenance. An appropriate evaluation method, one of the basic methods used to ensure the safe and reliable operation of the charging facility system. Another study [11] used fuzzy AHP to evaluate the electrical and safety performance of a charger. The characteristics of AHP are simple and systematic, but subjective, and the evaluation results lack objectivity. It is more reasonable to use the combined weights of AHP and entropy weight to calculate comprehensive weights [13,14]. The entropy weight method is an objective weighting method that is not influenced by the subjective random error of expert judgment on the weight result and can be combined with the analytic hierarchy method. The determined comprehensive weight obtained is also more reliable [13,14]. Another study [15] combined the entropy weight method with the AHP to determine the comprehensive weight. In Liu et al. [16], the analytic hierarchy process and the entropy weight method were used to combine subjective and objective evaluation methods to evaluate the operation of the distribution network and verify the rationality of the algorithm. In Yang et al. [17], the AHP and the entropy weight method were combined for the evaluation of the operating state of the charger. Other studies [18,19] combined the AHP and the entropy weight method to determine the combined weights, and then evaluated the objects to be evaluated.

In this study, we evaluated the safety level of the charging facility performance. Firstly, 18 indicators from three aspects were selected to construct an evaluation system for the health status of EV chargers based on the analysis of existing EV charging standards. Secondly, the analytic hierarchy process and entropy weight method were combined to determine the comprehensive weight of the index.

Then, the theory of fuzzy comprehensive evaluation was used to evaluate the performance of the charger. Finally, the above indicator system and evaluation method were used to evaluate a charger, demonstrating the effectiveness of the indicator system and the applicability of the evaluation method. According to the evaluation results, operation, and maintenance personnel can accurately and quickly diagnose and correct abnormal states or fault states of various charging piles to prevent or eliminate malfunction. Our method can provide necessary guidance for the operation of the equipment to improve the reliability, safety, and effectiveness of the equipment's' operation, and reduce the property loss to a minimum, which indirectly improves the performance of electric vehicles.

The contents of this paper are structured as follows: In Section 2, the evaluation index system is constructed; the processing methods of quantitative and qualitative index evaluation values are explained in Section 3. In Section 4, the theory and method of the comprehensive decision scoring applied are explained, including the analytic hierarchy process, entropy weight method, and fuzzy comprehensive evaluation theory. Section 5 provides an analysis of actual cases and describes the rationality and feasibility of verification methods. In Section 6, related analysis results of this work are presented.

## 2. Construction of the Index System

The construction of the electric vehicle charger index system is crucial for the efficient and reasonable evaluation of the state of the charger. The evaluation indicators of EV chargers mostly involve electrical performance, safety performance, and pile condition. For the evaluation of the state of EV chargers, there is no specific indicator system standard that can be referenced in China. As such, considering the impact on the safe and reliable operation of the charger, we included the principles of objectivity, system completeness, independence, and operability with the current charger test standard [20–22]. Considering the three reference planes of general performance, electrical performance, and safety performance, an index system suitable for the state evaluation of EV chargers was constructed, as outlined in Table 1.

**Table 1.** Performance status evaluation index system of electric vehicle (EV) charging infrastructure.

| Goal Layer | Criterion Layer | Index Layer |
|---|---|---|
| Electric vehicle charger evaluation index system (A0) | General Performance (B1) | Appearance performance (C11)<br>Ingress protection (IP) defense grade (C12)<br>Noise intensity (C13) |
| | Electrical Performance (B2) | Stable voltage accuracy (C21)<br>Stable current accuracy (C22)<br>Output-voltage setting error (C23)<br>Output-current setting error (C24)<br>Ripple factor (C25)<br>Efficiency (C26)<br>Power Factor (C27) |
| | Safety Performance (B3) | Input over-voltage protection (C31)<br>Input under-voltage protection (C32)<br>Output over-voltage protection (C33)<br>Output over-current protection (C34)<br>Over-voltage limitation characteristics (C35)<br>Over-current limitation characteristics (C36)<br>Security alarm (C37)<br>Insulation performance (C38) |

## 3. Method for Processing Indicator Evaluation Values

The performance indicators of the charger can be divided into a quantitative index and a qualitative index. Because the unit of each state indicator quantity is inconsistent, the charger cannot be evaluated using the original data, thus each indicator needs to be unified. In this paper, from the pros and cons of the evaluation indicators, the evaluation value of the indicators is uniformly treated using the percentage system. For the determination of the quantitative index scores, the score is constructed using the piece-wise function; for the qualitative indicators, the merits and demerits of the charger

indicators are scored according to the operation and maintenance personnel or expert experience. The higher the score, the better the operating state of the charger [23,24].

### 3.1. Quantitative Index

According to the corresponding function relationship, the function relationship between the precision value and the score value is established, and some index corresponding function relationships are listed as follows.

### 3.1.1. Voltage Stability Accuracy

In the NB/T 33001 standard, the voltage stability accuracy of the charger is specified as ±0.5%. In this paper, the index score and the precision value satisfy the linear function relationship $y = \pm40x + 100$. When the voltage stability accuracy is ±0.5%, the definition score is 80; when the voltage stability accuracy is ±1%, the definition score is 60.

### 3.1.2. Current Stability Accuracy

In the NB/T 33001 standard, the current stability accuracy of the charger is specified as ±0.5%. In this paper, the index score and the precision value satisfy the linear function relationship $y = \pm20x + 100$: When the current stability accuracy is ±1%, the definition score is 80; when the current stability accuracy is ±2%, the definition score is 60.

### 3.1.3. Ripple Coefficient

In the NB/T 33001 standard, the ripple coefficient of the charger is specified as ±0.5%. In this paper, the index score and the precision value satisfy the linear function relationship $y = \pm40x + 100$, meaning when the ripple coefficient is ±0.5%, the definition score is 80; when the ripple coefficient is ±1%, the definition score is 60.

### 3.1.4. Output Voltage Setting Error

In the NB/T 33001 standard, the output voltage setting error of the charger is specified as ±0.5%. In this paper, the index score and the precision value satisfy the linear function relationship $y = \pm40x + 100$: When the output voltage setting error is ±0.5%, the definition score is 80; when the output voltage setting error is ±1%, the definition score is 60.

### 3.1.5. Output Current Setting Error

In the NB/T 33001 standard, the output current setting error of the charger is specified as ±0.1%. In this paper, the index score and the precision value satisfy the linear function relationship $y = \pm20x + 100$. When the output current setting error is ±1%, the definition score is 80; when the output current setting error is ±2%, the definition score is 60.

### 3.2. Qualitative Indicators

Qualitative indicators were combined with the impact of indicators on the charger, using expert experience to score. The higher the score, the more stable the performance of the indicator. The input over-voltage protection performance score is as follows:

Input over-voltage protection:

(1) Act according to the specified protective conditions, score: 85;
(2) The protection action and the specified operating conditions are slightly deviated, or the action time is slightly delayed, score: 75;
(3) Protection action and movement time are obviously abnormal, score: 65.

## 4. Comprehensive Decision-Making Scoring Theory and Method

The abnormal operation state of an EV charger directly affects the power supply network, the power storage battery, and the charger itself, thus it is necessary to establish an evaluation model that can accurately and quickly find the charger problem. We considered the possible effects of various indicators of the charger. The proposed evaluation model can be summarized into two steps. The first step involves combining the AHP and entropy weight method to determine the comprehensive weight of the indicator; the second step is to use the fuzzy comprehensive evaluation method to score the charger [14].

### 4.1. Analytic Hierarchy Process

AHP is a decision analysis method for weight analysis of qualitative and quantitative indicators proposed by Thomas L. Saaty in the early 1970s [13,14]. The method of determining the subjective weight of the electric vehicle charger index by using the analytic hierarchy process is as follows:

(1) Determine the hierarchy model: In this paper, the evaluation index system constructed by the charger was divided into 3 levels: Target layer A, criterion layer B, and indicator layer C. The quasi-side layer contains B1, B2, and B3, and the index layer includes C11–C38.

(2) Structural judgment matrix: The judgment matrix was determined by the pairwise comparison of the factors between the factors of each layer. The elements in the matrix were determined using the 1–9 scale method.

(3) Calculated weight vector: The maximum eigenvalue of the constructed judgment matrix and the eigenvector corresponding to the largest eigenvalue were obtained, and then the weighted value of the required corresponding index was obtained by normalizing the obtained eigenvector.

(4) Consistency check: The maximum eigenvalue of each judgment matrix and its corresponding eigenvector were calculated, and the calculated consistency index, random consistency index, and consistency ratio were checked for consistency. If the requirement was met, the normalized feature vector was the weight vector. Otherwise, the elements in the judgment matrix needed to be adjusted, the judgment matrix was reconstructed, and the above steps were repeated until satisfied.

### 4.2. Entropy Weight Method

AHP is a subjective empowerment decision-making method, which cannot comprehensively consider objective factors. Conversely, the entropy weight method is an objective weighting method, and the combination of the two can compensate for the subjective randomness of AHP [13,14]. The step of determining the weight using the entropy weight method are as follows:

(1) Build the raw data matrix: Suppose there are $m$ evaluation objects, $n$ evaluation indicators, and the evaluation value of the $i$th evaluation object M$i$ for the $j$th index D$j$ is recorded as $x_{ij}$, and the formed original data matrix can be expressed as:

$$X = \begin{pmatrix} x_{11} & \cdots & x_{1n} \\ \vdots & \ddots & \vdots \\ x_{m1} & \cdots & x_{mn} \end{pmatrix} \tag{1}$$

(2) Calculate the feature weight of the $i$th evaluation object under the $j$th index. The feature weight of the $i$th evaluation object under the $j$th index is denoted as $p_{ij}$:

$$p_{ij} = x_{ij} / \sum_{i=1}^{m} x_{ij} \tag{2}$$

(3) Calculate the entropy weight $w_j$.

$$e_j = -k \sum_{i=1}^{m} p(x_{ij}) \ln p(x_{ij}) \tag{3}$$

$$w_j = 1 - e_j / \sum_{j=1}^{n} (1 - e_j) \tag{4}$$

where $k = 1/\ln(m)$, $k > 1$, and $0 \leq e_j \leq 1$.

(4) Determine the comprehensive weight A of the indicator:

$$a_j = \delta q_i + (1 - \delta) w_i \tag{5}$$

where $q_i$ is the subjective weight of the evaluation index, $w_i$ is the objective weight of the evaluation index, and $\delta$ is the preference coefficient of the subjective weight of the evaluation index, and $0 \leq \delta \leq 1$. According to the conclusion in the literature [23], the $\delta$ value is 0.5: The comprehensive weight of each indicator of the charger is the arithmetic mean of the objective weight and the subjective weight, which ensures the accuracy and reliability of the weight.

### 4.3. Fuzzy Comprehensive Evaluation

Fuzzy comprehensive evaluation is an evaluation method based on the theory of fuzzy mathematics and the comprehensive evaluation of the charger is achieved according to the principle of maximum membership [14]. In this paper, the fuzzy theory method is combined with AHP and entropy weight method to evaluate the state of the charger.

The specific evaluation steps are as follows:

(1) Determination of the comment set: We divided the comment set of the charger indicator status into fault, abnormal, attention, normal, and good into five levels. The comment set $V$ = {fault, abnormal, attention, normal, good} = {v1, v2, v3, v4, v5}.

(2) Determination of membership function: According to the evaluation index system established above, combined with the fuzzy set operation theory of the membership function in the literature [24,25], we combined the fuzzy distribution of a triangle and trapezoid, and established the membership function of the rating of the charger as follows:

$$\mu^1(x) = \begin{cases} 1 & x \geq 95 \\ \frac{1}{10}x - 8.5 & 85 \leq x < 95 \\ 0 & x < 85 \end{cases} \tag{6}$$

$$\mu^2(x) = \begin{cases} -\frac{1}{10}x + 9.5 & 85 \leq x \leq 95 \\ \frac{1}{10}x - 7.5 & 75 \leq x < 85 \\ 0 & x < 75 \end{cases} \tag{7}$$

$$\mu^3(x) = \begin{cases} -\frac{1}{10}x + 8.5 & 75 \leq x \leq 85 \\ \frac{1}{10}x - 6.5 & 65 \leq x < 75 \\ 0 & x < 65 \end{cases} \tag{8}$$

$$\mu^4(x) = \begin{cases} -\frac{1}{10}x + 7.5 & 65 \leq x \leq 75 \\ \frac{1}{10}x - 5.5 & 55 \leq x < 65 \\ 0 & x < 55 \end{cases} \tag{9}$$

$$\mu^5(x) = \begin{cases} 1 & x < 50 \\ -\frac{1}{15}x + \frac{13}{3} & 50 \leq x < 65 \\ 0 & x \geq 65 \end{cases} \tag{10}$$

where $\mu^k(x)$ means the membership function of the comment $k$; and $k = \{1, 2, 3, 4, 5\}$, respectively, indicate the five levels of good, normal, attention, abnormal, and faulty. The score range of the comment set with different indicators is: Fault: 0–60, abnormal: 60–70, attention: 70–80, normal: 80–90, and good: 90–100.

(3) Determination of the scoring matrix $R$: After converting the raw data of each indicator into corresponding scores, they were brought into the corresponding membership function to perform a single factor evaluation, and a single factor evaluation set of corresponding indicators was obtained. The matrix formed by the single factor evaluation of each indicator is recorded as the scoring matrix $R$.

(4) Fuzzy comprehensive evaluation: On the basis that the scoring matrix $R$ and the comprehensive weight A have been determined, the scoring matrix composed of each index was calculated by $B = A \times R$, and the fuzzy comprehensive evaluation result vector $B$ of each index could be obtained. Similarly, the total fuzzy comprehensive evaluation result vector of the charger was $B_T$.

(5) Calculate comprehensive score: The fuzzy comprehensive evaluation result was brought into the membership function for defuzzification calculation, and the score of the comment set was $V_T = (v1, v2, v3, v4, v5)$. Calculated using Equation (11), the comprehensive score $F$ of the charger was obtained, where $F$ can be described as:

$$F = \frac{B_T \times V_T}{\sum B_T} \tag{11}$$

The basic process of performance evaluation of electric vehicle charger is shown in Figure 1.

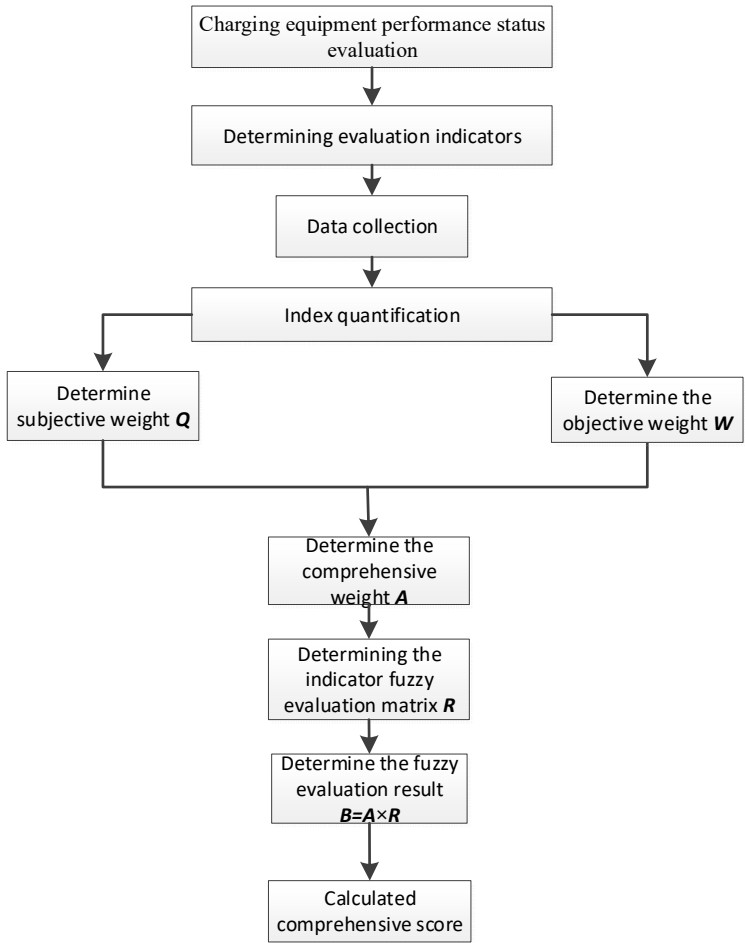

**Figure 1.** Charging infrastructure evaluation flow chart.

## 5. Analysis of Examples

A charger made by a manufacturer in Shijiazhuang was selected and evaluated with the test data of the charger test platform.

(1) Calculation of the subjective weight: Combining the characteristics of the work of the charger and comprehensive expert opinions, the comprehensive weight of each indicator was obtained.

The judgment matrix and subjective weight of the criterion layer are shown in Table 2. The maximum eigenvalue is 3.0246, and the random consistency ratio (CR) is 0.0236, which is less than 0.1, which satisfies the requirement of consistency.

**Table 2.** Subjective weight of the rule hierarchy.

| Index | B1 | B2 | B3 | Q0 |
|-------|----|----|----|------|
| B1 | 1 | 1/4 | 1/5 | 0.0974 |
| B2 | 4 | 1 | 1/2 | 0.3331 |
| B3 | 5 | 2 | 1 | 0.5695 |

The subjective weight judgment matrix of general performance is shown in Table 3, in which the maximum eigenvalue is 3.0536, and the random CR is 0.0516 (<0.1), which satisfies the consistency requirement.

**Table 3.** Subjective weight of the general performance.

| Index | C11 | C12 | C13 | Q1 |
|-------|-----|-----|-----|------|
| C11 | 1 | 1/2 | 1/2 | 0.1958 |
| C12 | 2 | 1 | 1/2 | 0.3108 |
| C13 | 2 | 2 | 1 | 0.4934 |

The subjective weight judgment matrix of electrical performance is shown in Table 4, in which the maximum eigenvalue is 7.6051, and the random CR is 0.0742 (<0.1), which satisfies the consistency requirement.

**Table 4.** Subjective weight of the electrical performance.

| Index | C21 | C22 | C23 | C24 | C25 | C26 | C27 | Q2 |
|-------|-----|-----|-----|-----|-----|-----|-----|------|
| C21 | 1 | 1 | 1/2 | 1/2 | 3 | 3 | 3 | 0.1351 |
| C22 | 1 | 1 | 1/2 | 1/2 | 3 | 3 | 3 | 0.1351 |
| C23 | 2 | 2 | 1 | 1 | 6 | 8 | 8 | 0.2865 |
| C24 | 2 | 2 | 1 | 1 | 6 | 8 | 8 | 0.2865 |
| C25 | 1/3 | 1/3 | 1/6 | 1/6 | 1 | 7 | 7 | 0.0943 |
| C26 | 1/3 | 1/3 | 1/8 | 1/8 | 1/7 | 1 | 1 | 0.0313 |
| C27 | 1/3 | 1/3 | 1/8 | 1/8 | 1/7 | 1 | 1 | 0.0313 |

The subjective judgment matrix of safety performance is shown in Table 5, where the maximum eigenvalue is 8.5962, and the random CR is 0.0604 (<0.1), which satisfies the consistency requirement.

(2) The calculation of objective weights: The objective weights were calculated according to the test data recorded by the test platform at different time periods, the field operation, maintenance personnel, and expert experience, which is represented by W.

For the scores of general performance, electrical performance, and safety performance in the criterion layer, the average score of each sub-indicator was taken and then the objective weight of the index was obtained according to the entropy weight method as shown in Tables 6–8, respectively.

**Table 5.** Subjective weight of the safety performance.

| Index | C31 | C32 | C33 | C34 | C35 | C36 | C37 | C38 | Q3 |
|---|---|---|---|---|---|---|---|---|---|
| C31 | 1 | 1 | 1 | 1 | 1/3 | 1/3 | 2 | 1/4 | 0.0648 |
| C32 | 1 | 1 | 1 | 1 | 1/3 | 1/3 | 2 | 1/4 | 0.0648 |
| C33 | 1 | 1 | 1 | 1 | 1/3 | 1/3 | 2 | 1/4 | 0.0648 |
| C34 | 1 | 1 | 1 | 1 | 1/3 | 1/3 | 2 | 1/4 | 0.0648 |
| C35 | 3 | 3 | 3 | 3 | 1 | 1 | 3 | 1/4 | 0.1580 |
| C36 | 3 | 3 | 3 | 3 | 3 | 1 | 3 | 1/4 | 0.1947 |
| C37 | 1/2 | 1/2 | 1/2 | 1/2 | 1/3 | 1/3 | 1 | 1/3 | 0.0475 |
| C38 | 4 | 4 | 4 | 4 | 4 | 4 | 3 | 1 | 0.3409 |

**Table 6.** Objective weight of the general performance.

| Index | C11 | C12 | C13 |
|---|---|---|---|
| | 85 | 85 | 80 |
| | 85 | 85 | 80 |
| Score | 85 | 85 | 80 |
| | 85 | 85 | 80 |
| | 85 | 85 | 80 |
| W | 0.3333 | 0.3333 | 0.3333 |

**Table 7.** Objective weight of the electrical performance.

| Index | C21 | C22 | C23 | C24 | C25 | C26 | C27 |
|---|---|---|---|---|---|---|---|
| Standard limited value | ±0.5% | ±1% | ±1% | ±1% | ±1% | 93 | 0.98 |
| | 0.255/89.80 | 0.669/86.62 | 1.697/32.12 | 0.735/85.30 | 0.474/81.04 | 92.926/79.88 | 0.974/78.50 |
| | 0.207/91.72 | 0.572/88.56 | 1.718/31.28 | 1.434/71.32 | 0.426/82.96 | 93.322/80.92 | 0.991/91.00 |
| Measured values (%)/score | 0.22/91.20 | 0.493/90.14 | 1.701/31.96 | 1.56/68.80 | 0.453/81.88 | 93.350/80.97 | 0.997/97.00 |
| | 0.212/91.52 | 0.349/93.02 | 1.699/32.04 | 1.735/65.30 | 0.435/82.60 | 93.679/81.94 | 0.996/96.00 |
| | 0.22/91.20 | 0.321/93.58 | 1.786/28.56 | 1.593/68.14 | 0.41/83.60 | 93.787/82.22 | 0.998/98.00 |
| W | 0.0029 | 0.0449 | 0.1008 | 0.4883 | 0.0061 | 0.0055 | 0.3515 |

**Table 8.** Objective weight of the safety performance.

| Index | C31 | C32 | C33 | C34 | C35 | C36 | C37 | C38 |
|---|---|---|---|---|---|---|---|---|
| | 85 | 85 | 85 | 85 | 85 | 85 | 85 | 80 |
| | 85 | 85 | 85 | 85 | 85 | 85 | 85 | 80 |
| Score | 85 | 85 | 85 | 85 | 85 | 85 | 85 | 80 |
| | 85 | 85 | 75 | 85 | 75 | 85 | 85 | 80 |
| | 85 | 85 | 75 | 85 | 75 | 85 | 85 | 80 |
| W | 0.0000 | 0.0000 | 0.5000 | 0.0000 | 0.5000 | 0.0000 | 0.0000 | 0.0000 |

(3) Comprehensive weight calculation.

On the basis of calculating the objective weight 'W' and the subjective weight 'Q', the comprehensive weight A is calculated according to the conclusion in the literature [23], the comprehensive weight of each indicator of the charger is the arithmetic mean of the objective weight and the subjective weight, which is shown in Table 9.

(4) Fuzzy Comprehensive Evaluation: Combine the B1, B2, and B3 obtained by obscuring the general performance, electrical performance, and safety performance to form a matrix $R_T$ of the fuzzy evaluation of the total performance of the charger.

Use $B_T = A_T \times R_T$ = (0.0725 0.1152 0.2762 0.4810 0.0095) to derive the state of membership of the charger, and then calculate the BT anti-fuzzification to obtain a review set $V_T$ for the state of the charger: $V_T$ = (63.3665 56.7290 67.5810 78.8370 85.2145).

**Table 9.** Comprehensive weight of the indicators.

| Weight\Index | General Performance | | | Criterion Layer | | |
|---|---|---|---|---|---|---|
| | **C11** | **C12** | **C13** | **B1** | **B2** | **B3** |
| W | 0.3333 | 0.3333 | 0.3333 | 0.0000 | 0.2308 | 0.7692 |
| Q | 0.1958 | 0.3108 | 0.4934 | 0.0974 | 0.3331 | 0.5695 |
| A | 0.2646 | 0.3221 | 0.4134 | 0.0487 | 0.2819 | 0.6694 |

| Weight\Index | Electrical Performance | | | | | | |
|---|---|---|---|---|---|---|---|
| | **C21** | **C22** | **C23** | **C24** | **C25** | **C26** | **C27** |
| W | 0.0029 | 0.0449 | 0.1008 | 0.4883 | 0.0061 | 0.0055 | 0.3515 |
| Q | 0.1351 | 0.1351 | 0.2865 | 0.2865 | 0.0943 | 0.0312 | 0.0312 |
| A | 0.0690 | 0.0900 | 0.1936 | 0.3874 | 0.0502 | 0.0184 | 0.1914 |

| Weight\Index | Safety Performance | | | | | | | |
|---|---|---|---|---|---|---|---|---|
| | **C31** | **C32** | **C33** | **C34** | **C35** | **C36** | **C37** | **C38** |
| W | 0.0000 | 0.0000 | 0.5000 | 0.0000 | 0.5000 | 0.0000 | 0.0000 | 0.0000 |
| Q | 0.0648 | 0.0648 | 0.0648 | 0.0648 | 0.1579 | 0.1947 | 0.0475 | 0.3408 |
| A | 0.0324 | 0.0324 | 0.2824 | 0.0324 | 0.3290 | 0.0973 | 0.0237 | 0.1704 |

Finally, according to Equation (11), the charger state score, $F = 71.09$ points, was obtained. The state of the charger was classified as "attention" and was close to the abnormal state. Combined with the analysis of the original data, the index layer determines that the output voltage tuning error has reached the fault value and the output current tuning error has reached an abnormal value, and other straight indicators are normal. The test results of the abnormal indicators obtained using the test platform based on Microsoft Visual Studio 2013 is shown in Figures 2 and 3.

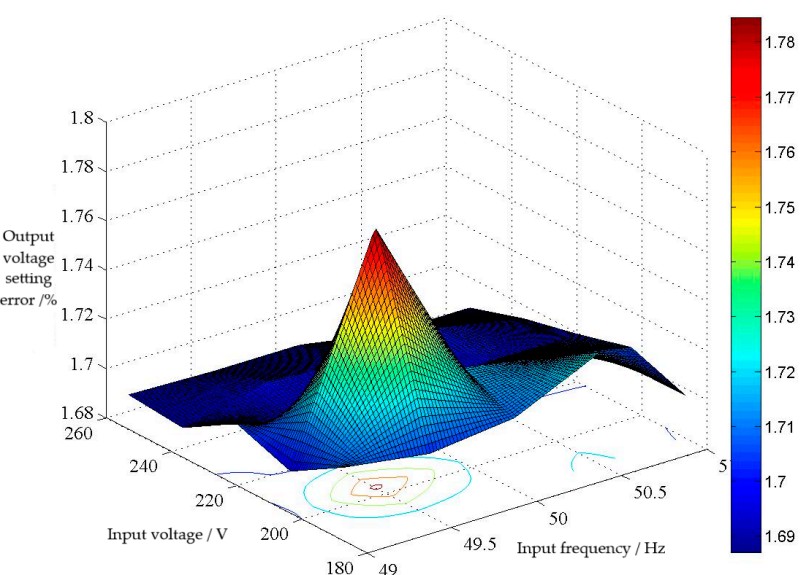

**Figure 2.** Output voltage setting error test result diagram.

Figure 2 depicts the test result of the output voltage setting error when the input voltage and input frequency were changed by the standard, which is obviously higher than ±0.5% of the standard limit value. Figure 3 shows the output current setting error. The test results when the standard specified output voltage and output current range change are obviously higher than the standard limit value by ±1%. The above result shows that the obtained charger evaluation result is consistent with the test report, which verifies the effectiveness of the evaluation model and the reliability of the indicator system.

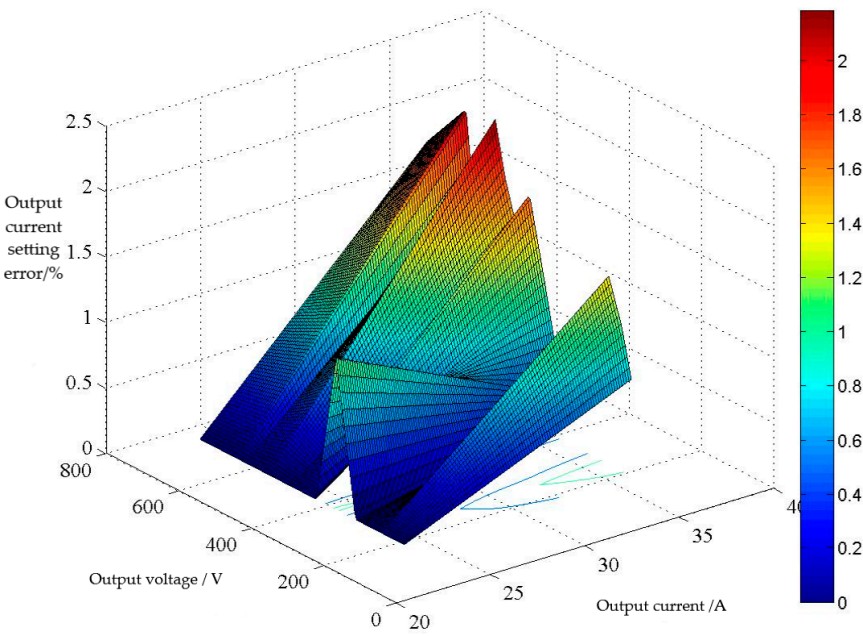

**Figure 3.** Output current setting error test result diagram.

## 6. Conclusions

As charging accidents occur occasionally, the safety of EV charging equipment has become increasingly important. We studied the evaluation of the performance of an EV charger. We selected 18 indicators from three aspects to form an evaluation index system for the performance status of EV chargers. Based on these, a fuzzy comprehensive evaluation method was used to evaluate an electric vehicle charger, which quantified the performance status of the charging device. With the detection of EV chargers, the continuous improvement in safety standards, and the continuous accumulation of on-site operation and maintenance data, a fault experience library can be formed, providing a basis for the improvement of on-site operation and maintenance of EV chargers. Based on the evaluation results, operation and maintenance personnel can have a comprehensive understanding of the health of the tested charger. According to the evaluation results, operation and maintenance personnel can quickly and accurately diagnose and correct the abnormal or fault states of various charging piles to prevent or eliminate a malfunction. Our method can provide necessary guidance for the operation of the equipment to improve its reliability, safety, and effectiveness, and reduce the property loss to a minimum, which indirectly improves the performance of electric vehicles.

**Author Contributions:** Conceptualization, Q.L. and Y.X.; methodology, S.Z. and, X.Z.; software, Z.L. and L.W.; validation, S.Z., X.Z., Z.L., and L.W.; formal analysis, X.Y.; investigation, Z.W. and X.Z.; data curation, S.Z.; writing—original draft preparation, S.Z. and Y.W.; writing—review and editing, Q.L., Y.X., and X.Y.

**Funding:** This research was supported by the project "AC/DC charging interface interoperability testing and fault characterization analysis technology for on-site inspection of electric vehicle charging facilities (9210118009)".

**Conflicts of Interest:** The authors declare no conflict of interest. The sponsors had no role in the design, execution, interpretation, or writing of the study.

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
