# Peer review of "Performance Status Evaluation of an Electric Vehicle Charging Infrastructure Based on the Fuzzy Comprehensive Evaluation Method"

_wevj, doi:10.3390/wevj10020035_

Round 1
Reviewer 1 Report
In their article, the authors address the condition assessment of charging infrastructure. In principle, the paper is written in a comprehensible way, but some points should be added or improved before a publication can be recommended.
First of all, some punctuation errors and missing or incorrectly positioned spaces are noticeable. This already starts in the abstract (e.g. lines 18, 22 and 24) and continues. Therefore a spellcheck should be done again.
My next point of criticism concerns the rather limited state of research. This should definitely be deepened and extended.
Finally, the authors should better work out why it makes sense to reduce all indicators on the state of the charge point to one key indicator. This seems counter-intuitive at first, since only the individual indicators for maintenance provide information about concrete shortcomings and conditions. For example, would it not also make sense to stay on the criterion layer, i.e. to monitor three indicators per attachment point, in order to be able to specify immediately which layer has which state? I hope I am understandable: it is not a question of describing the work of the authors as not relevant, but of the authors working out the motivation and the necessity of the work better.
The authors should also add in the discussion how the work fits into the state of research, where it is supplemented and where limitations may still exist.
Author Response
Point 1: First of all, some punctuation errors and missing or incorrectly positioned spaces are noticeable. This already starts in the abstract (e.g. lines 18, 22 and 24) and continues. Therefore a spellcheck should be done again.
Response 1: some punctuation errors and missing or incorrectly positioned spaces has been corrected.
Point 2: My next point of criticism concerns the rather limited state of research. This should definitely be deepened and extended.
Response 2: The state of the equipment is extended by giving the test results of anomaly index.
Point 3: The authors should better work out why it makes sense to reduce all indicators on the state of the charge point to one key indicator. This seems counter-intuitive at first, since only the individual indicators for maintenance provide information about concrete shortcomings and conditions.
Response 3: Finally, in this study, the state of the charging equipment is taken as the research target, and the indicators are divided according to the functional differences and properties of different indicators. The criterion layer is the embodiment of the performance and function differences of the indicators, and is prepared for the subsequent hierarchical positioning of faults. In addition, because the evaluation results obtained by the evaluation method used in this paper are processed and evaluated according to the data measured by the actual indicator status, when the charging device fails or is abnormal, the evaluation result according to the fuzzy evaluation can be obtained;then, according to the maximum membership principle based on fuzzy comprehensive evaluation, fault location of the specific fault or abnormality of the charging device is performed, thereby accurately determining the index of one or more faults, and providing a reliable basis for equipment operation and maintenance.

Reviewer 2 Report
The topic is interesting and it is in congruence with the mission of the World Electric Vehicle Journal. The subject may attract interest to the readers. In general, this manuscript is well written, with detailing the framework approach of the study and clearly stated methodology.
- Please add more references. In the Introduction Section, please provide more general information on the importance of research in order to emphasize the state of the art also (first general information, then specific). Also, references should be provided for the methods used in the work. Analytic Hierarchy Process, entropy weight method, fuzzy theory method are not personal contributions of the authors.
The manuscript could be improved by provide more information, details of the results presented in the tables.
- Few more specific comments and recommendations:
Please check the format of text and make sure it corresponds to the template.
Define all notations that is used where the concept appears first mentioned in the main text.
In section 2. Materials and Methods ... L79 Table 1 - link the table to the main text. Also, Table 6.
Usually, conclusions are supported by the analysis or by the discussions included in the main text of the paper. Please provide and highlight relevant aspects of your work so that they contain 3 to 5 short bullet points that convey the essential conclusions of your article.
There are some typos in the manuscript. Please double-check.
Having mentioned the above, this manuscript is proposed to be published after minor revision (corrections to minor errors and text editing).
Author Response
Point 1: Please add more references. In the Introduction Section, please provide more general information on the importance of research in order to emphasize the state of the art also (first general information, then specific). Also, references should be provided for the methods used in the work. Analytic Hierarchy Process, entropy weight method, fuzzy theory method are not personal contributions of the authors.
Response 1: More relevant references have been added to the introduction, and more references have been added to the methods used. For detailed changes, see the attached Word file.
For details of the modification, please refer to the attached Word file.
Point 2: Usually, conclusions are supported by the analysis or by the discussions included in the main text of the paper. Please provide and highlight relevant aspects of your work so that they contain 3 to 5 short bullet points that convey the essential conclusions of your article.
Response 2: At the end of the paper, the result graph of the anomaly indicator is added to better explain the conclusion, and at the end, the conclusion part is revised and the main points are refined.
For details of the modification, please refer to the attached Word file.
